# Impact of moisturizing pretreatments on small reusable dental instruments cleanliness and mechanics

Xiuyu Tang[1], Yi Min[1,2,3], Yixuan Geng[1], Huan Huang[1], Tianjuan Xia[1,2,3]*

1 State Key Laboratory of Oral & Maxillofacial Reconstruction and Regeneration, Wuhan, Hubei, P.R.China, 2 Key Laboratory of Oral Biomedicine Ministry of Education, Wuhan, Hubei, P.R.China, 3 Hubei Key Laboratory of Stomatology, School and Hospital of Stomatology, Wuhan University, Wuhan, Hubei, P.R.China

◉ These authors contributed equally to this work.
* 1035597145@qq.com

## Abstract

### Background

Effective cleaning of reusable dental instruments is crucial. Pre-cleaning moisturizing aims to prevent bioburden from drying; however, its comparative efficacy and impact on instrument integrity require further investigation.

### Aim

This *in vitro* study evaluated the effects of various moisturizing pretreatments on the cleaning efficacy and mechanical properties of small reusable dental instruments.

### Methods

Surgical burs (fissure, round) and Nickel-Titanium (NiTi) files (n = 30/type) underwent four cycles of use and reprocessing with one of three pretreatments: water (control), a multi-enzyme cleaner, or a professional moisturizer (instrument transport gel). Cleanliness was assessed by microscopy (25 × magnification) and Adenosine Tri-phosphate (ATP) testing. Scanning Electron Microscope (SEM) and Energy Dispersive X-ray Spectroscopy (EDS) were used to evaluate surface contamination and wear after cycling.

### Results

The professional moisturizer yielded significantly higher cleanliness scores and lower ATP values across all instrument types compared to the water or multi-enzyme cleaner groups (P < 0.05), maintaining consistent cleaning efficacy. SEM revealed less surface contaminant residue in the professional moisturizer group. However, SEM also indicated more frequent structural alterations on burs (fissure and round) pretreated

**Data availability statement:** All relevant data are within the manuscript and its Supporting Information files.

**Funding:** "This research was financially supported by the Wuhan University Clinical Nursing Special Research and Cultivation Fund Project (1607/600400021). The funders had no role in study design, data collection and analysis, decision to publish, or preparation of the manuscript. There was no additional external funding received for this study".

**Competing interests:** The authors declare that they have no known competing financial interests or personal relationships that could have appeared to influence the work reported in this paper.

with the professional moisturizer. EDS analysis suggested better preservation of the instruments' native elemental composition in the professional moisturizer group.

## Conclusions

Professional moisturizer pretreatment significantly improved cleaning and reduced bioburden on reusable dental instruments. Although superior in cleaning, the observed structural alterations on burs pretreated with the professional moisturizer warrant further investigation into its long-term effects on specific instrument material integrity before broad recommendations can be made for extending instrument lifespan.

---

## 1. Introduction

Small reusable dental instruments, such as cutting burs, nickel-titanium (NiTi) files, and implant instruments, are characterized by their diverse types, complex structures, varied materials, and often significant cost. These instruments are designed to cut, abrade, or otherwise interact with dental structures and materials. To ensure patient safety and prevent cross-infection in clinical settings, thorough cleaning, disinfection, and sterilization are essential before each reuse. The quality of cleaning is a critical prerequisite for effective subsequent disinfection and sterilization [1]. However, the irregular surface morphology of these instruments, combined with contamination from tissue debris, blood, saliva, and residual dental materials during use, presents significant cleaning challenges [2]. If instruments are not cleaned promptly after use, adhered organic matter can dry, making subsequent cleaning substantially more difficult. Inadequate cleaning compromises the efficacy of disinfection and sterilization, potentially leading to sterilization failure [3]. Furthermore, prolonged contact with organic substances, such as blood, can corrode instruments and shorten their operational lifespan [4]. Consequently, numerous guidelines and standards recommend that reusable instruments undergo pretreatment, such as moisturization, promptly after use and before definitive cleaning [5,6].

Moisturization pretreatment is considered crucial for optimizing subsequent cleaning, disinfection, and sterilization processes. However, existing research on moisturization pretreatments has often been limited to visual assessment of surface cleanliness and has not consistently employed more rigorous, quantitative methodologies. Furthermore, there is a paucity of data regarding the long-term impact of different pretreatment methods on residual surface contamination and the mechanical integrity of reusable dental instruments subjected to multiple reprocessing cycles [7,8].

This in vitro study, therefore, aimed to evaluate and compare three distinct moisturization pretreatment methods--water, an enzyme-containing cleaner, and a professional instrument transport gel (hereafter referred to as 'professional moisturizer') -- applied to commonly used reusable dental instruments. The primary objective was to identify a pretreatment method that not only enhances initial cleaning efficacy and reduces bioburden but also maintains these effects consistently across multiple reuse cycles while minimizing detrimental impacts on the instruments' mechanical properties.

## 2. Materials and methods

### 2.1. Ethical statement

The experimental protocol was reviewed and approved by the Ethics Committee of the School and Hospital of Stomatology, Wuhan University (Approval No.WDKQ2025（B32）)

### 2.2. Materials

The following materials were used: Surgical high-speed long fissure burs (Walrus Dental Instrument Co., Ltd., Foshan, China), NiTi M3 root canal instruments (Shanghai Yirui Dental Materials Co., Ltd., Shanghai, China), Slow-speed caries removal round burs (Walrus Dental Instrument Co., Ltd., Foshan, China), Operating microscope with integrated full HD camera (1080 P60) (Model OMS2350; Zumax Medical Co., Ltd., Suzhou, China), a Scanning Electron Microscope(SEM) (Model MIRA3; TESCAN ORSAY HOLDING, a.s., Brno, Czech Republic), an ATP luminometer（Biotech-LD100; Beijing Chuang Xin Shi Ji Science & Technology Development Co.Ltd., Beijing, China), ATP sampling swabs (UltraSnap™; Hygiena LLC, Camarillo, CA, USA), Instrument transport Gel (PRE-Klenz™; STERIS Corporation, Mentor, OH, USA)–referred to as "professional moisturizer," a Multi-enzyme cleaner (WEIGERT neodisher® MultiZym; Chemische Fabrik Dr. Weigert GmbH & Co. KG, Hamburg, Germany) and Sterile deionized water (for control group).

### 2.3. Study design and experimental groups

A total of 90 new dental instruments were used, comprising three types: surgical high-speed long fissure burs (n = 30), slow-speed caries-removing round burs (n = 30), and NiTi M3 root canal instruments (n = 30). For each instrument type, the 30 instruments were randomly allocated into three pretreatment groups (n = 10 instruments per instrument type per group):

Control Group (Water): Instruments were pretreated by immersion in sterile deionized water.

Multi-enzyme cleaner Group (Enzyme): Instruments were pretreated by immersion in the multi-enzyme cleaner, prepared according to the manufacturer#39;s instructions.

Professional Moisturizer Group (Moisturizer): Instruments were pretreated by applying instrument transport gel according to the manufacturer#39;s instructions.

Each instrument underwent four cycles of clinical use (contamination), pretreatment, cleaning, and assessment.

The specific group designations were as follows:

Fissure Burs:

WF Group (n = 10): Water pretreatment

EF Group (n = 10): Multi-enzyme cleaner pretreatment

MF Group (n = 10): Professional moisturizer pretreatment

Round Burs:

WR Group (n = 10): Water pretreatment

ER Group (n = 10): Multi-enzyme cleaner pretreatment

MR Group (n = 10): Professional moisturizer pretreatment

NiTi Instruments:

W-NiTi Group (n = 10): Water pretreatment

E-NiTi Group (n = 10): Multi-enzyme cleaner pretreatment

M-NiTi Group (n = 10): Professional moisturizer pretreatment

## 2.4. Clinical use (contamination) and pretreatment protocol

Before the first cycle and after each assessment in subsequent cycles, all instruments were contaminated by performing a standardized cutting procedure on clinical human teeth for 60 seconds. Then, the instruments were allowed to air-dry for 30 minutes at room temperature (approximately 22–25°C) to allow debris to adhere. Immediately after contamination and drying, instruments were assigned to their respective pretreatment groups. Instruments in the Water and Multi-enzyme cleaner groups were fully immersed in their respective solutions. Instruments in the Professional Moisturizer group were coated with the gel. All pretreatments were carried out for 2 hours at room temperature in covered containers to prevent drying of the pretreatment agent. Following the 2-hour pretreatment, instruments were removed by a trained operator using sterile forceps, processed in an ultrasonic bath with neutral detergent for 15 minutes and then thoroughly rinsed.

## 2.5. Cleanliness assessment: optical microscopy and scoring

After each cleaning step in every cycle, instrument cleanliness was evaluated under the operating microscope (OMS2350) at 25× magnification. Digital images were captured using the integrated camera before contamination (baseline for new instruments) and after each cleaning procedure. Two independent examiners, blinded to the pretreatment group, evaluated the coded images. Cleanliness was scored using the following criteria: 0 points: Obvious stains, visible debris, and/or blood stains present. 1 point: Two of the following present – slight stains, minor debris, or trace blood. 2 points: One of the following presents – a very slight stain, minimal debris, or a minute trace of blood. 3 points: No visible stains, debris, or blood detected. Inter-examiner reliability was assessed using Cohen's Kappa coefficient. If discrepancies occurred, a consensus was reached by joint re-evaluation.

## 2.6. Bioburden quantification: Adenosine Triphosphate (ATP) bioluminescence assay

Immediately following microscopy assessment in each cycle, the entire working surface of each instrument was swabbed using an UltraSnap™ ATP sampling swab according to the manufacturer's instructions. The swab was then inserted into the ATP luminometer to measure the level of residual ATP, recorded in Relative Light Units (RLU). Higher RLU values indicate greater residual organic contamination.

## 2.7. Surface integrity and elemental analysis: Scanning Electron Microscope (SEM) and Energy Dispersive X-ray Spectroscopy (EDS)

After the completion of all four use and reprocessing cycles, representative instruments from each group (n = 3 per subgroup) were prepared for SEM and EDS analysis.

**SEM:** Instruments were cleaned with ethanol, air-dried, and sputter-coated with gold-palladium. Surface morphology, including the integrity of cutting edges, presence of fractures, or wear patterns, was observed using the MIRA3 SEM. Images were acquired in high-vacuum, secondary electron detection mode at an electron acceleration voltage of 20 kV, with magnifications ranging from 200× to 10,000×.

**EDS:** Elemental composition analysis of selected areas on the instrument surfaces (particularly areas showing debris or wear) was performed using an X-MaxN EDS detector (Oxford Instruments, Abingdon, UK) coupled to the SEM. This was used to identify the nature of any residual contaminants or changes in the instrument material.

## 2.8. Statistical analysis

Data were analyzed using SPSS version 21.0 (IBM Corp., Armonk, NY, USA), and graphs were generated using Microsoft Excel 2019 (Microsoft Corp., Redmond, WA, USA).

 

Microscopy scores (ordinal data) were analyzed using the Kruskal-Wallis test, followed by Mann-Whitney U tests with Bonferroni correction for pairwise comparisons between groups at each cycle. The Friedman test was used for within-group changes across cycles.

ATP (RLU values) data were first tested for normality using Shapiro-Wilk test. If the distribution is normal, repeated measures ANOVA or mixed-effects model is used to compare groups across cycles, followed by post-hoc tests such as Tukey's or Bonferroni. If the distribution is not normal, the Friedman test for within-group changes and Kruskal-Wallis for between-group comparisons at each cycle is used, followed by Mann-Whitney U tests with Bonferroni correction. A P-value < 0.05 was considered statistically significant. Inter-examiner reliability for microscopy scores was calculated using Cohen's Kappa Coefficient.

## 3. Results

### 3.1. Cleanliness scores and ATP bioluminescence detection

The calculated Cohen's Kappa value for inter-examiner reliability was 0.87, indicating substantial agreement. The cleanliness and residual microbial/organic load on instruments were assessed using optical microscopy and ATP bioluminescence, respectively, after each of the four cycles of simulated use and reprocessing.

**3.1.1. Surgical high-speed long fissure burs.** For surgical high-speed long fissure burs, instruments pretreated with the professional moisturizer consistently exhibited higher cleanliness scores compared to those treated with the multi-enzyme cleaner or water across all four cycles (Fig 1A). Similarly, ATP bioluminescence values, indicating microbial/ organic residue, were significantly lower for the professional moisturizer group compared to both the water and multi-enzyme cleaner groups (P < 0.05) (Fig 1B). No significant differences in cleanliness scores or ATP RLU values were observed between the water-treated and multi-enzyme cleaner-treated groups for this instrument type. The professional moisturizer group demonstrated consistent cleaning performance throughout the repeated use and reprocessing cycles.

**3.1.2. Slow-speed round burs.** Slow-speed round burs pretreated with the professional moisturizer also showed higher cleanliness scores and significantly lower ATP RLU values compared to the multi-enzyme cleaner and water groups in the initial cycles (P < 0.05) (Figs 2A,2B). The professional moisturizer group generally maintained better cleanliness over the cycles. However, by the fourth reuse cycle, the differences in cleanliness scores and ATP RLU values between the professional moisturizer group and the multi-enzyme cleaner group diminished, suggesting a more comparable performance between these two pretreatments after extended use for this specific type of bur.

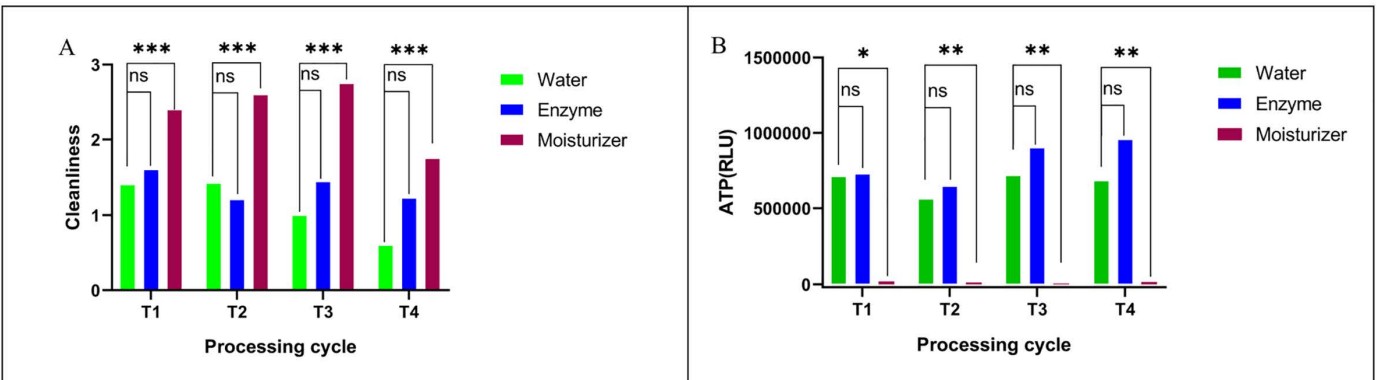

**Fig 1. Effect of different moisturizing pretreatments on (A) cleanliness scores and (B)surface microbial residues (ATP RLU values) of surgical high-speed long fissure burs (mean ± SD).** (T1, T2, T3, and T4 indicate the cycle of reuse. Statistical significance: * P < 0.05; ** P < 0.01; ***P < 0.001; ns: P > 0.05 (not significant).

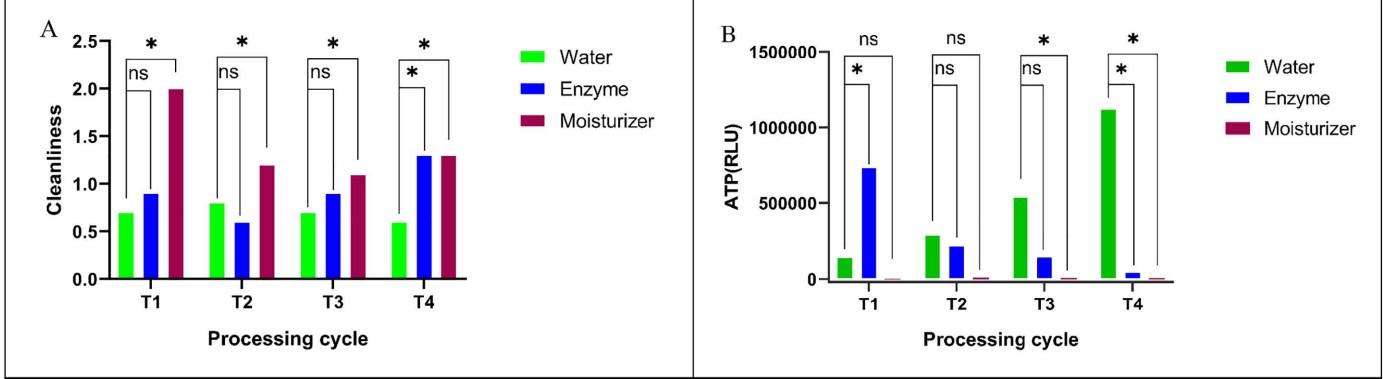

**Fig 2. Effects of different moisturizing pretreatments on (A) cleanliness scores and (B) surface microbial residues (ATP RLU values) of slow-speed round burs (mean±SD).** (T1, T2, T3, and T4 indicate the cycle of reuse. Statistical significance: * P<0.05; ns, P>0.05 (not significant).

**3.1.3. Nickel-Titanium (Ni-Ti) root canal instruments.** For Ni-Ti root canal instruments, pretreatment with the professional moisturizer resulted in higher cleanliness scores (Fig 3A) and significantly lower ATP RLU values (P<0.05) (Fig 3B) when compared to both water and multi-enzyme cleaner pretreatments across all cycles. The professional moisturizer was effective in maintaining the cleanliness of these instruments throughout the use and reprocessing cycles.

## 3.2. SEM analysis of surface condition

SEM was used to qualitatively assess surface contamination and structural integrity of the instruments after four cycles of use and reprocessing.

**3.2.1. Instruments moisturized with water.** Following four cycles, fissure burs (WF group) and round burs (WR group) pretreated with water generally exhibited substantial microbial attachment and adhered debris on their surfaces. Ni-Ti instruments (W-NiTi group) pretreated with water also showed surface contamination, although qualitatively less than observed on the burs. Significant structural damage, including loss of integrity of cutting blades, was evident on the fissure and round burs in this group (Fig 4).

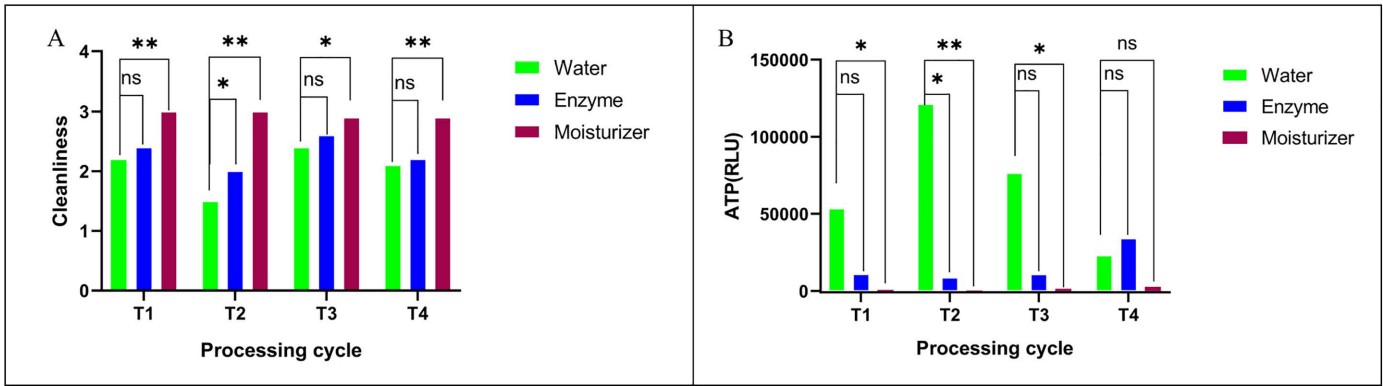

**Fig 3. Effect of different moisturizing pretreatments on (A) cleanliness scores and (B) surface microbial residues (ATP RLU values) of Ni-Ti root canal instruments (mean±SD).** (T1, T2, T3, and T4 indicate the cycle of reuse. Statistical significance: * P<0.05; ** P<0.01; ns, P>0.05 (not significant.).

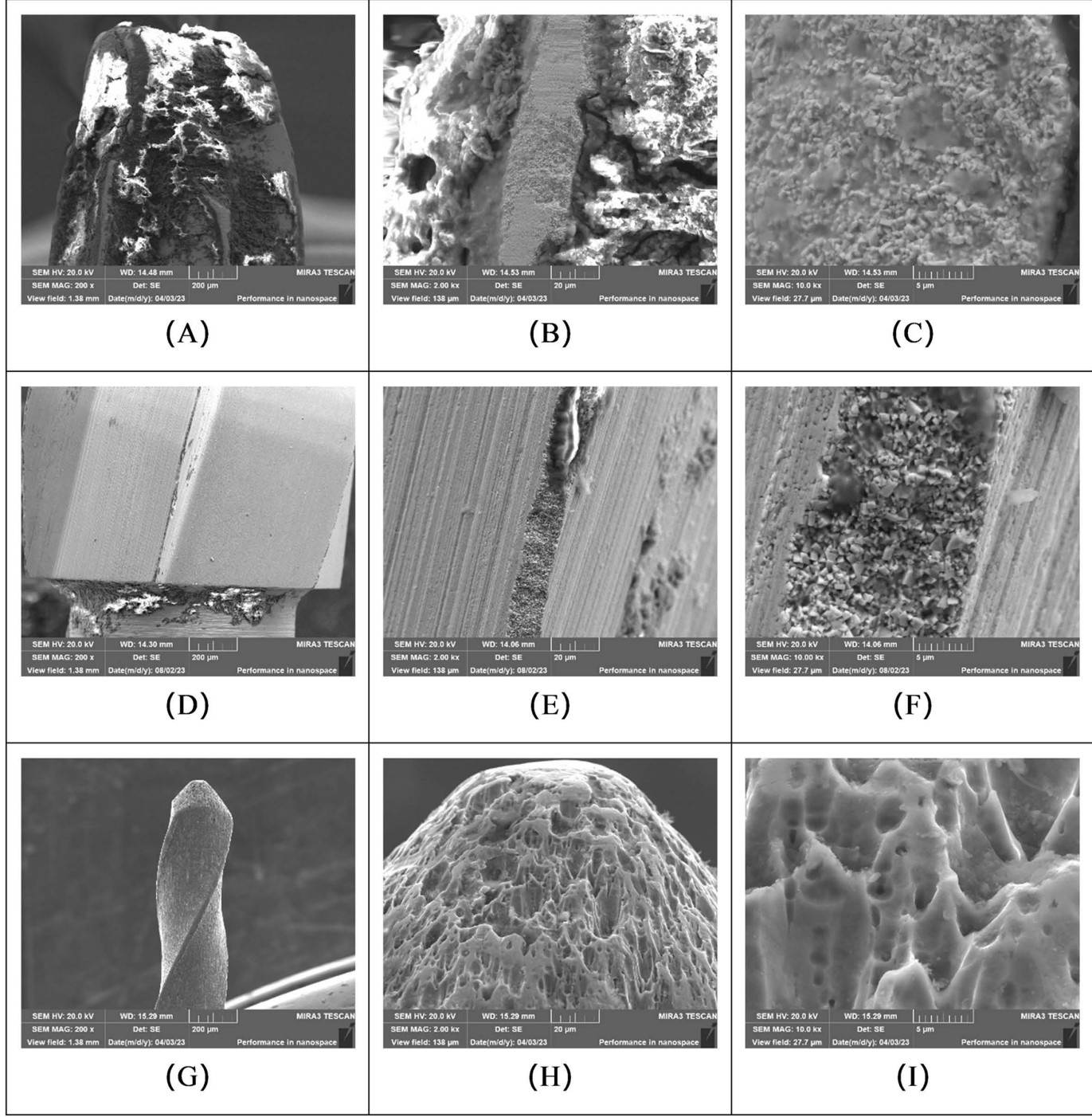

**Fig 4. SEM micrographs of instruments pretreated with water, showing microbial attachment and structural wear after four reuse cycles.**
Representative images from: **(A-C)** Surgical high-speed long fissure bur (WF group); **(D-F)** Slow-speed round bur (WR group); and **(G-I)** Ni-Ti root canal instrument (W-NiTi group). **(A)** Fissure bur at 200 × magnification, indicating substantial accumulation of debris. **(B)** Fissure bur at 2000x magnification, showing structural damage and debris. **(C)** Fissure bur at 10000 × magnification, detail of an area filled with debris. **(D)** Round bur at 200 × magnification, indicating the presence of debris. **(E)** Round bur at 2000 × magnification, showing structural damage with partial areas filled with debris. **(F)** Round bur at 10000 × magnification, detail of damage and debris. **(G)** Ni-Ti instrument at 200 × magnification, showing relatively little debris on the main shaft. **(H, I)** Active tip of the Ni-Ti instrument at 2000× and 10000 × magnification, respectively, showing obvious structural damage.

**3.2.2. Instruments pretreated with Multi-enzyme Cleaner.** Instruments pretreated with the multi-enzyme cleaner (EF, ER, and E-NiTi groups) also displayed microbial attachments and adhered debris to their surfaces after four cycles. Obvious structural damage, particularly loss of integrity of the cutting blades, was observed on the surgical high-speed long fissure burs and slow-speed round burs (Fig 5).

**3.2.3. Instruments pretreated with professional moisturizer.** SEM images of instruments pretreated with the professional moisturizer (MF, MR, M-NiTi groups) showed comparatively less microbial attachment and adhere debris to their surfaces (Fig 6). However, observations of the instrument structure indicated that wear, such as alterations to cutting edges or surface integrity, was more commonly noted in the fissure bur (MF) and round bur (MR) groups pretreated with the professional moisturizer compared to baseline conditions or other forms of wear seen in other groups.

## 3.3. Energy Dispersive X-ray Spectroscopy (EDS) analysis of surface composition

EDS analysis was performed on selected areas of instrument surfaces after four cycles to identify the elemental composition of visible residues and the underlying instrument material.

Overall, EDS findings suggested that pretreatment with the professional moisturizer assisted in preserving the native surface elemental composition of the instruments with less evidence of embedded biological or extrinsic material compared to the other groups.

Detailed EDS analysis of a representative fissure bur (Fig 7), presumably from a group with significant visible contamination (water treated), revealed that areas heavily laden with debris showed high levels of phosphorus and calcium, indicative of residual dental tissue or biological matter (Fig 7 area1, area3, and area4). Uncovered areas of the same bur confirmed the base material as Tungsten (Fig 7 area2).

Further EDS analysis of a tungsten bur surface (Fig 8), potentially exhibiting some remaining particulate matter, confirmed its primary composition as tungsten (W). Similarly, EDS analysis of a Ni-Ti instrument surface (Fig 9) confirmed its composition as Nickel (Ni) and Titanium (Ti) in areas clear of gross debris. These findings (Figs 8, 9) illustrate the ability of EDS to identify the instrument#39;s base material composition, which, when less obscured by contaminants (as was generally the case for the professional moisturizer group as noted by SEM), suggests better maintenance of the original surface.

## 4. Discussion

The effective reprocessing of small, reusable dental instruments, characterized by their complex structures and diverse materials [9,10], is a persistent challenge in dental practice. These instruments frequently become contaminated with patient-derived materials such as blood, tissue debris, and microorganisms, which can rapidly dry on their surfaces if not addressed promptly, thereby impeding subsequent cleaning and sterilization [4,11–14]. This underscores the necessity for effective pretreatment strategies, including moisturization, immediately after clinical use. However, a consensus on optimal moisturization protocols is lacking. This study aimed to address this by comparing three common pretreatment methods—water, an enzyme-containing cleaner, and a professional moisturizer—for their impact on the cleaning efficacy and surface integrity of surgical burs and Ni-Ti root canal instruments, which are frequently used in dental clinics.

Our study reinforces its utility in evaluating dental instrument reprocessing, as evidenced by higher optical microscopy cleanliness scores and significantly lower ATP bioluminescence values (indicating less organic/microbial residue), compared to water or multi-enzyme cleaner pretreatments across most instrument types and cycles. This aligns with previous research suggesting that dedicated moisturizing agents can be more effective than water alone or even some enzyme solutions in preventing the dedication of bioburden and facilitating its removal [15,16]. The use of an operating microscope at 25×magnification in this study, an advancement over the more common 5×magnification visual checks [13], likely provided a more sensitive assessment of surface cleanliness. While ATP bioluminescence is a well-established quantitative

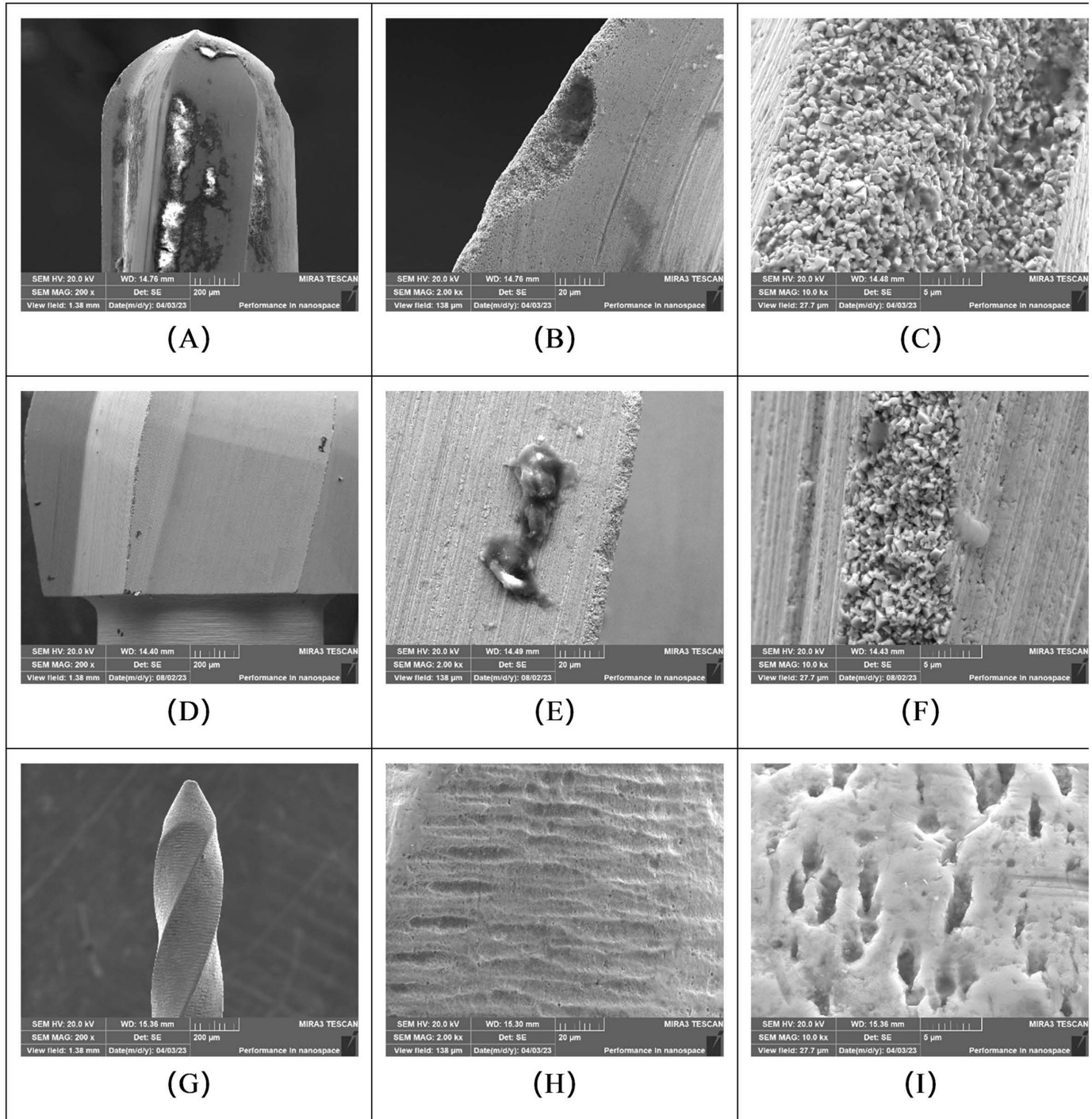

**Fig 5. SEM micrographs of instruments pretreated with multi-enzyme cleaner, showing microbial attachment and structural wear after four reuse cycles.** Representative images from: **(A-C)** Surgical high-speed long fissure bur (EF group); **(D-F)** Slow-speed round bur (ER group); and **(G-I)** Ni-Ti root canal instrument (E-NiTi group). **(A)** Fissure bur surface at 200×magnification, with some adhered debris. **(B)** Fissure bur at 2000×magnification, structural damage is present. **(C)** Fissure bur at 10000×magnification, detail of structural damage. **(D)** Round bur at 200×magnification, no obvious gross debris. **(E)** Round bur at 2000×magnification, showing structural damage and some debris. **(F)** Round bur at 10000×magnification, detail of damage and debris. **(G, H, I)** Ni-Ti instrument at 200×, 2000×, and 10000×magnification, respectively, showing relatively little debris and minor structural damage.

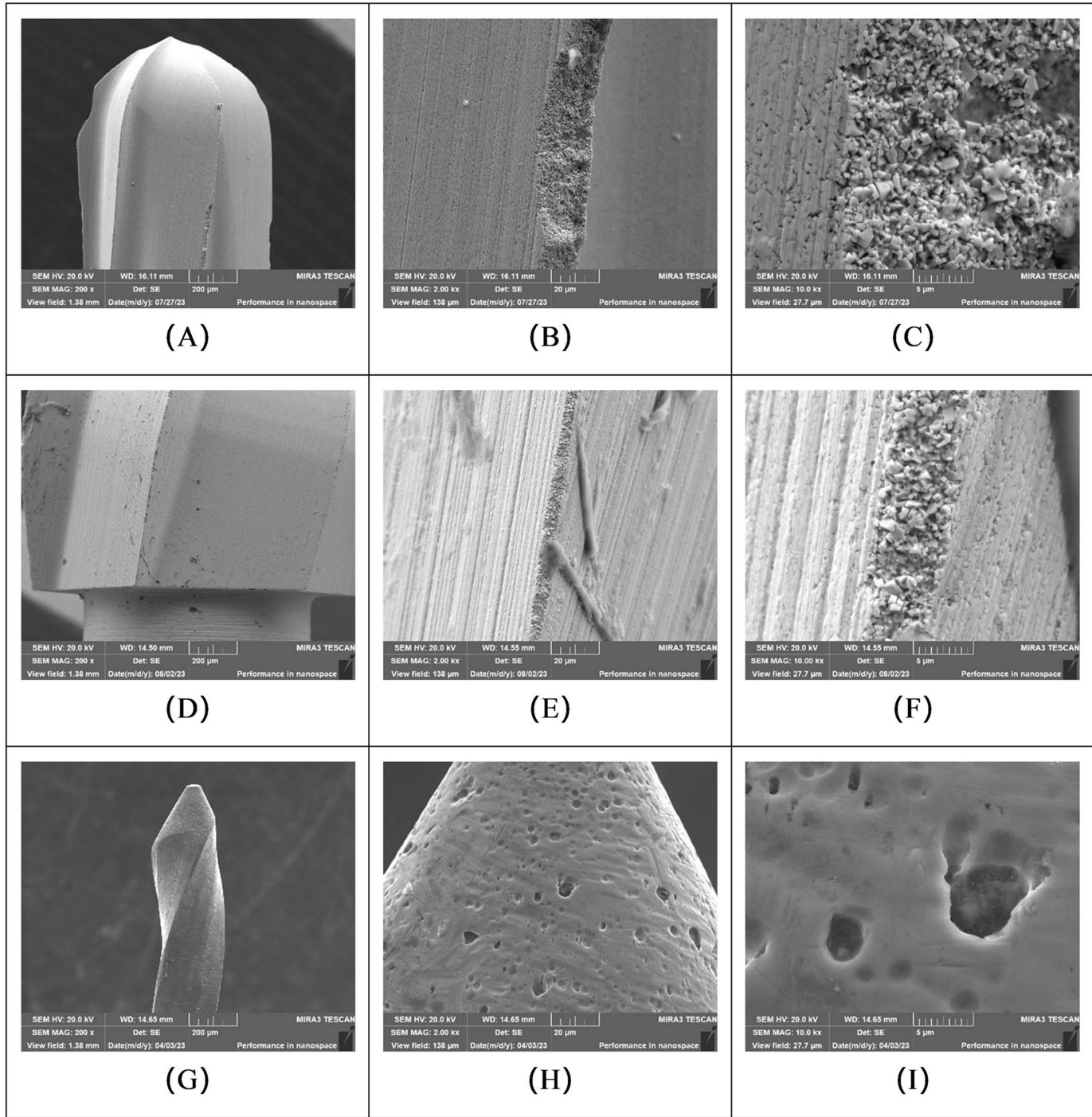

**Fig 6. SEM micrographs of instruments pretreated with professional moisturizer, showing microbial attachment and structural wear after four reuse cycles.** Representative images from: **(A-C)** Surgical high-speed long fissure bur (MF group); **(D-F)** Slow-speed round bur (MR group); and **(G-I)** Ni-Ti root canal instrument (M-NiTi group). **(A)** Fissure bur surface at 200 × magnification, with little adhered debris but some structural damage visible. **(B)** Fissure bur at 2000 × magnification, detail of structural damage. **(C)** Fissure bur at 10000 × magnification, further detail of obvious structural damage. **(D)** Round bur at 200 × magnification, no obvious gross damage. **(E)** Round bur at 2000 × magnification, showing some structural damage and minimal debris. **(F)** Round bur at 10000 × magnification, detail of structural damage. **(G, H, I)** Ni-Ti instrument at 200 ×, 2000 ×, and 10000 × magnification, respectively, showing very little debris and minimal structural damage.

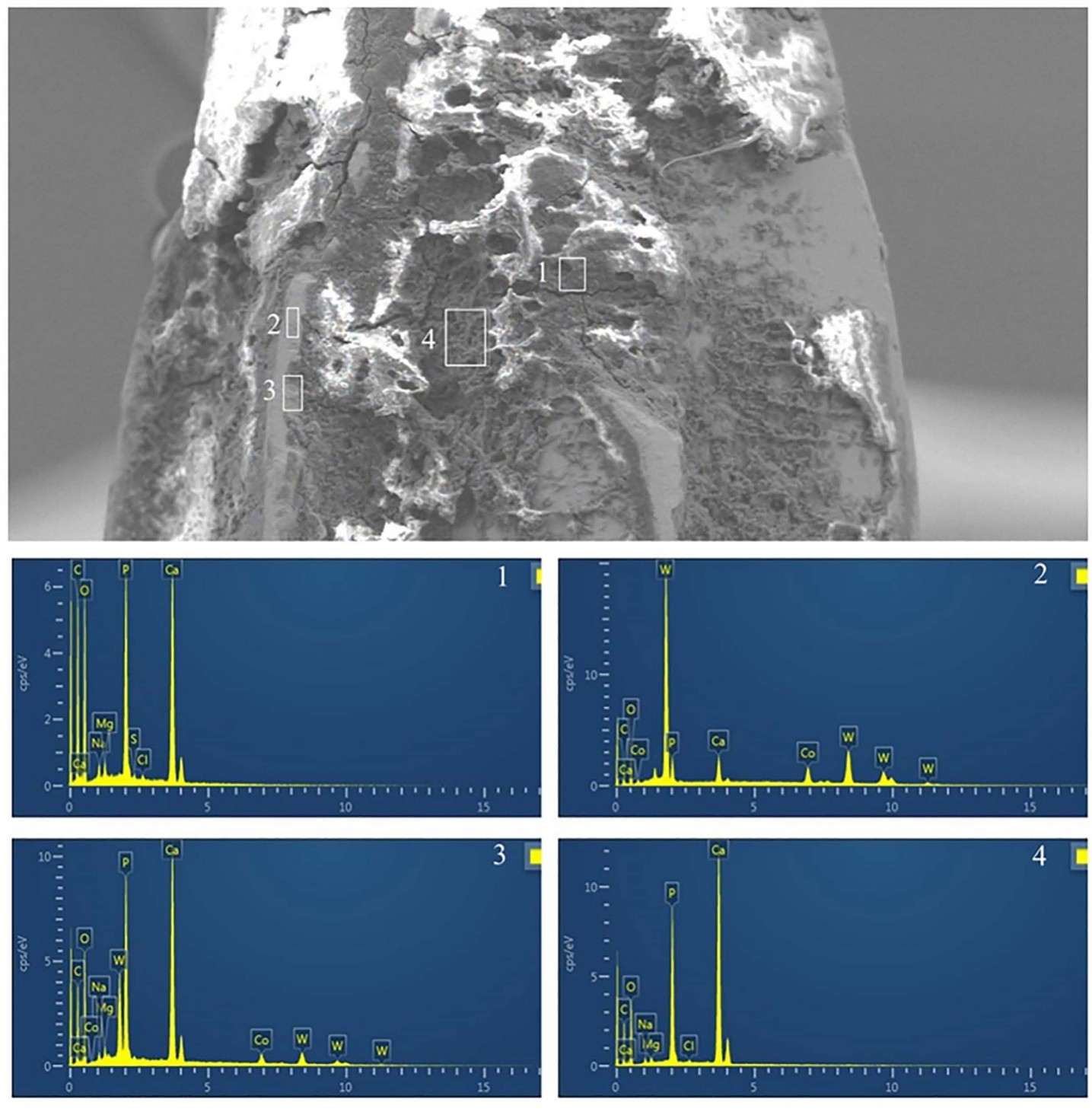

**Fig 7. EDS analysis of adhered debris on a representative surgical high-speed long fissure bur from the WF group (water pretreatment) after four reuse cycles.** The main image shows an SEM micrograph with numbered areas (1-4) selected for EDS. Accompanying graphs/spectra display the elemental composition for each selected area. (Key elements: Si, Silicon; W, tungsten; Al, aluminum; Au, gold (likely from sputter coating for SEM); C, carbon; Ca, calcium; Co, cobalt; Fe, iron; Mg, magnesium; Na, sodium; Ni, nickel; O, oxygen; P, phosphorus). High levels of Ca and P in debris areas (e.g., areas 1, 3, 4) suggest biological residues.

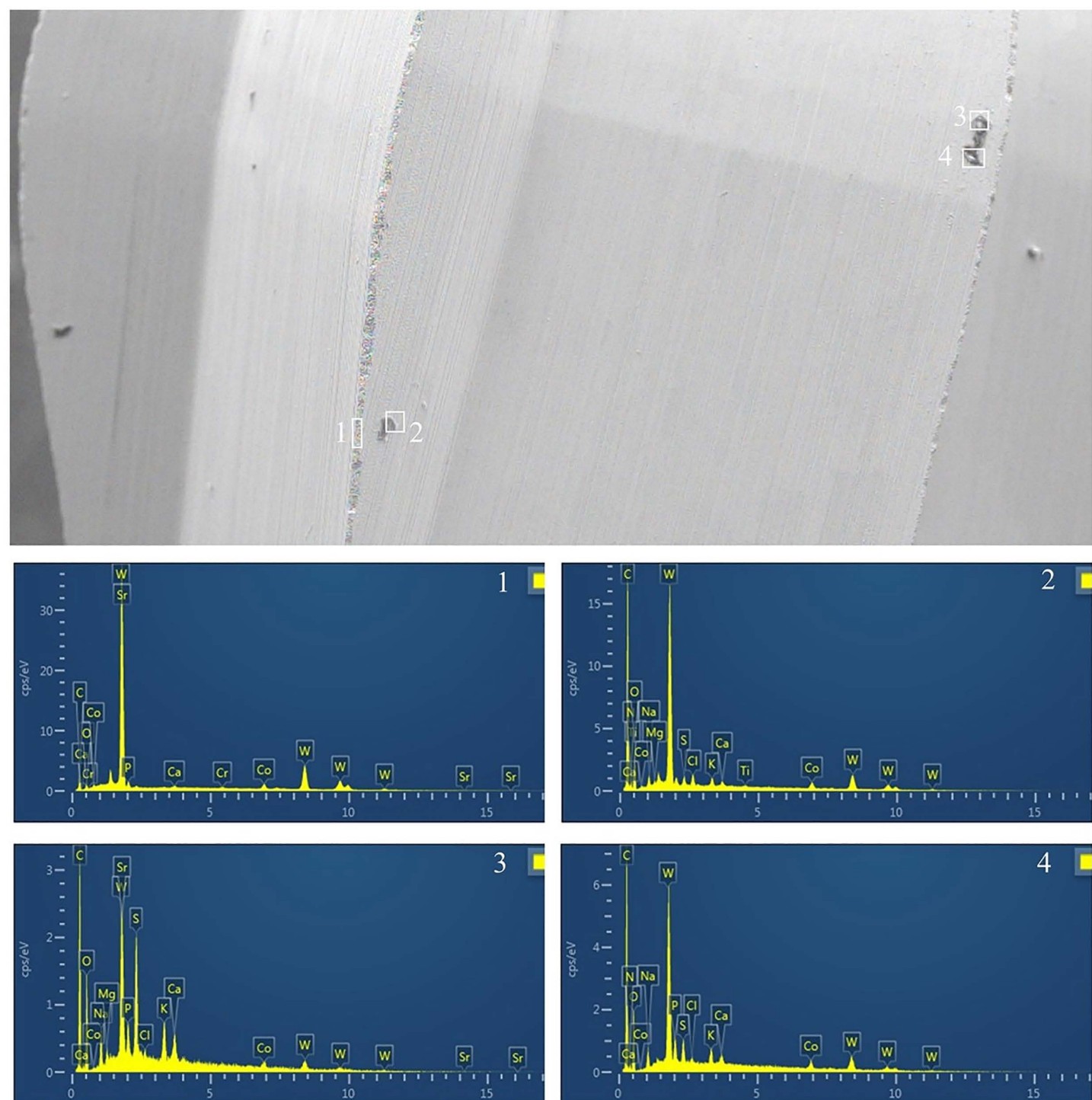

**Fig 8. EDS analysis of the surface of a representative slow-speed round bur from the MF group (professional moisturizer pretreatment) after four reuse cycles.** The main image shows an SEM micrograph with numbered areas (1-4) selected for EDS. Accompanying graphs/spectra display the elemental composition for each selected area. (Key elements: Si, Silicon; W, tungsten; Al, aluminum; Au, gold (likely from sputter coating for SEM); C, carbon; Ca, calcium; Co, cobalt; Fe, iron; Mg, magnesium; Na, sodium; Ni, nickel; O, oxygen; P, phosphorus). Analysis indicates the primary composition of tungsten (W) with minimal biological indicators in cleaner areas.

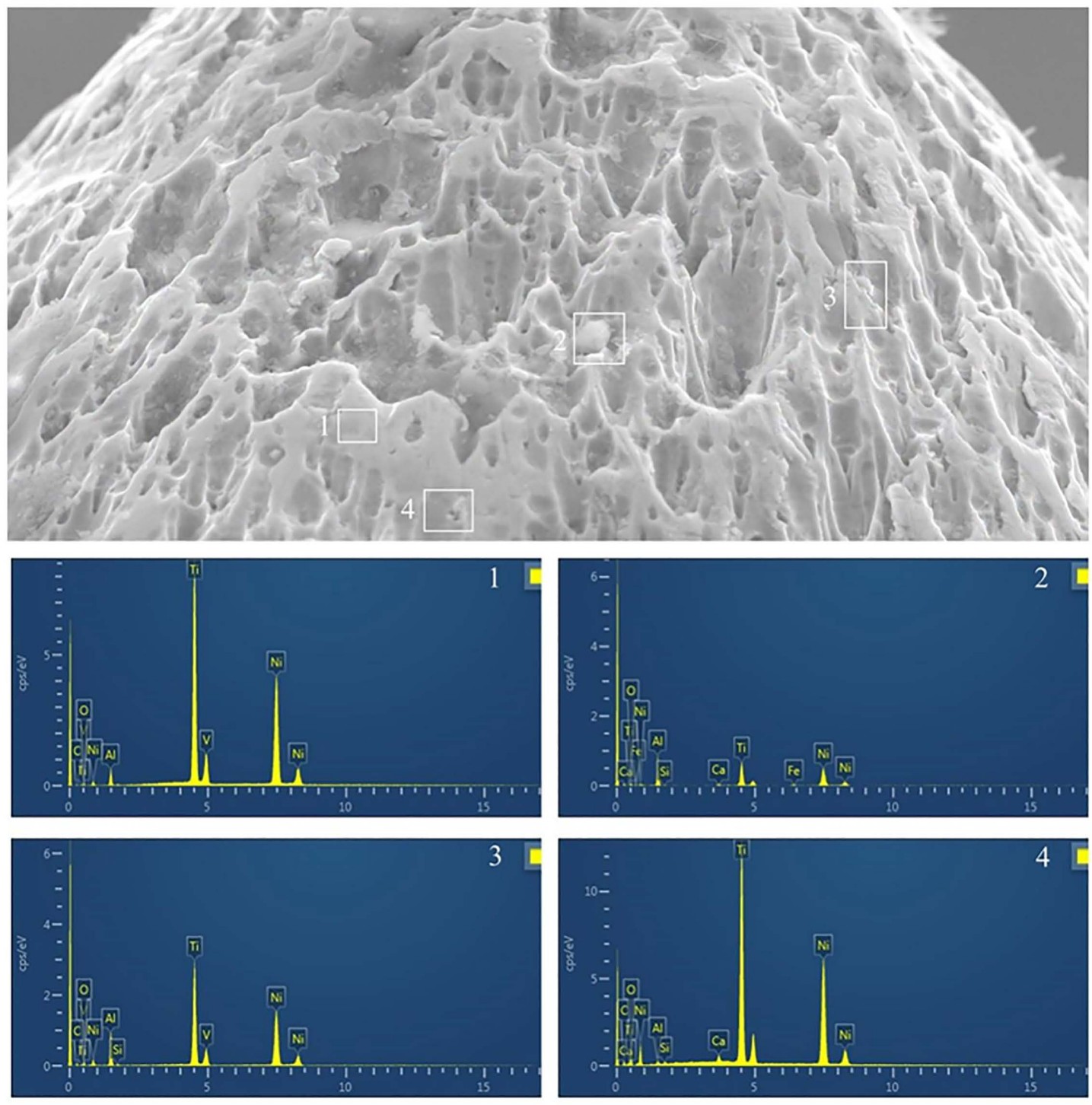

**Fig 9. EDS analysis of the surface of a representative Ni-Ti root canal instrument from the W-NiTi group (water pretreatment) after four reuse cycles.** The main image shows an SEM micrograph with numbered areas (1-4) selected for EDS. Accompanying graphs/spectra display the elemental composition for each selected area. (Key elements: Si, Silicon; W, tungsten; Al, aluminum; Au, gold (likely from sputter coating for SEM); C, carbon; Ca, calcium; Co, cobalt; Fe, iron; Mg, magnesium; Na, sodium; Ni, nickel; O, oxygen; P, phosphorus). Analysis confirms Ni and Ti as primary components, with the presence of other elements indicating surface contaminants.

method for assessing surface hygiene by detecting cellular ATP [17–21], our study reinforces its utility in evaluating the dental instrument reprocessing.

The enhanced efficacy of the professional moisturizer can be attributed to its multi-component formulation, which includes surfactants, chelating agents, and humectants. Unlike water, which is a passive solvent, or enzymatic cleaners that target specific organic molecules (e.g., proteases degrading proteins), this formulation provides a more comprehensive mechanism of action. Surfactants reduce the surface tension of the bioburden, enabling more effective dislodgement and solubilization. Concurrently, humectants prevent the desiccation of clinical soils into a tenacious, biofilm-like matrix, while chelating agents sequester the metal ions that stabilize this matrix [22,23]. This multifaceted approach is more robust in preventing the initial, tenacious adhesion of complex clinical debris compared to the more targeted, and potentially slower action of multi-enzyme cleaners.

These results are consistent with and expand upon the current literature. Walsh et al. also found that a moisturizing spray significantly reduced protein bioburden on endodontic instruments versus a delayed-cleaning control, though their study did not assess material integrity [24]. Conversely, the observation that the multi-enzyme detergent#39;s efficacy is highly dependent on contact duration and temperature [25] may explain its initially inferior performance in our study compared to the rapid action of the moisturizing agent. Although the surface degradation of the tungsten carbide burs was not anticipated, this finding aligns with material science principles, indicating that chronic exposure to specific chemical agents can alter the binder phase of cemented carbides, potentially compromising their mechanical properties [26,27]. The novelty of our investigation lies in its concurrent assessment of both cleaning performance and substrate integrity, providing a more comprehensive evaluation of these widely used pretreatment protocols.

For surgical high-speed long fissure burs and Ni-Ti instruments, the professional moisturizer consistently outperformed the other agents. Interestingly, for slow-speed round burs, while the professional moisturizer was initially superior, its performance in terms of cleanliness scores and ATP reduction became more comparable to that of the multi-enzyme cleaner by the fourth reprocessing cycle. This might suggest that for certain bur types or debris compositions, the cumulative effect of enzymatic action over repeated cycles can approach that of a dedicated moisturizer, or perhaps the nature of wear on round burs influences debris retention differently over time.

A significant aspect of this study was the investigation of instrument surface integrity after repeated reprocessing cycles, an area that was explored less in previous moisturization studies [28]. SEM analysis revealed that all instruments, regardless of pretreatment, exhibited some degree of surface contamination and structural wear after four cycles. Although instruments pretreated with the professional moisturizer generally showed less adhered debris and microbial attachment on their surfaces according to SEM, a noteworthy observation was that surgical burs (fissure and round types) from this group (MF and MR) exhibited what appeared to be more frequent or distinct structural alterations or damage compared to those seen in the other groups. This finding presents an apparent contrast, as one might expect cleaner surfaces to correlate with better preservation. It is possible that the highly effective cleaning action of the professional moisturizer, by thoroughly removing bioburden, exposed the underlying instrument surface more directly to mechanical stresses during handling and subsequent cleaning steps, or it may have revealed pre-existing micro-damage more clearly. Alternatively, the formulation of the professional moisturizer itself, while excellent for cleaning, might have a different chemical interaction with the bur materials over repeated exposures compared to the enzyme solution or water, leading to a different surface wear pattern, a phenomenon that warrants consideration, considering studies on chemical-material interactions during decontamination [29]. This specific observation warrants further investigation.

EDS analysis complemented the SEM findings by providing elemental information about surface residues and the underlying instrument material. The detection of phosphorus and calcium in debris on heavily contaminated instruments confirmed the presence of biological material, consistent with clinical contamination. The EDS data from instruments pretreated with the professional moisturizer generally showed a clearer representation of the instruments' native elemental composition (tungsten for burs, Ni and Ti for endodontic files) with less overlying elemental evidence of biological residue.

This suggests that while SEM might reveal certain types of surface alterations, the professional moisturizer was more effective at reducing the adhered biomaterial at an elemental level, which is critical for mitigating risks associated with retained organic soil. The apparent discrepancy between more visible structural alterations (via SEM) and better elemental surface cleanliness (via EDS) for the professional moisturizer group highlights the complexity of defining "instrument damage" and "wear." The observed "damage" in the professional moisturizer (PM) group may not necessarily equate to the greater loss of instrument material, but rather to a different surface texture or appearance.

The finding that all instruments showed some residual contamination and structural wear after only four cycles supports concerns raised in the literature regarding the long-term reusability and complete cleanability of such intricate devices. Although the ideal of single-use for all such instruments is often challenged by economic considerations in clinical practice, our results emphasize the need for meticulous reprocessing. Our study suggests that professional moisturizers offer a significant advantage in terms of initial and sustained cleaning efficacy and bioburden reduction. However, the observation of more frequent structural alterations on burs with the professional moisturizer necessitates careful consideration. It may indicate that although hygienically superior, instruments pretreated in this way should be monitored closely for wear, or the nature of this wear needs to be better understood regarding its clinical impact on cutting efficiency and fracture resistance.

This study has several limitations. As an *in vitro* investigation, it may not perfectly replicate the diverse conditions and contaminants encountered in a clinical setting. The specific standardized soil and the defined post-pretreatment cleaning protocol would also influence outcomes. The study was limited to four reprocessing cycles, and longer-term effects remain to be explored.

Future research should focus on elucidating the precise nature and clinical significance of the structural alterations observed on burs pretreated with professional moisturizers. Longer-term studies with a higher number of cycles, diverse instrument brands, and varying clinical soil loads are warranted. Furthermore, investigating the impact of different subsequent manual or automated cleaning protocols following various pretreatments would provide valuable insights.

In conclusion, pretreatment with a professional moisturizer significantly enhanced the cleaning efficacy and reduced the bioburden on small reusable dental instruments compared to water or the tested multi-enzyme cleaner. Although this method appears beneficial for achieving superior hygiene, the associated observations of structural alterations on buses warrant further investigation to fully understand the balance between cleaning efficacy and material integrity for these agents. Nevertheless, based on the critical importance of minimizing bioburden, professional moisturizers demonstrate considerable promise as a pretreatment step in the reprocessing of dental instruments.

## Supporting information

**S1 Data. Experimental data and analysis results.**
(XLSX)

## Acknowledgments

The authors thank Liangyan Li for providing professional training and guidance on the operation of the ATP detection equipment and Chao Wang for participating in some data analysis.

## Author contributions

**Conceptualization:** Tianjuan Xia.

**Formal analysis:** Xiuyu Tang, Yi Min, Yixuan Geng, Huan Huang.

**Funding acquisition:** Tianjuan Xia.

**Investigation:** Xiuyu Tang, Yi Min, Yixuan Geng, Huan Huang.

**Methodology:** Xiuyu Tang.

**Project administration:** Tianjuan Xia.

**Supervision:** Tianjuan Xia.

**Writing – original draft:** Xiuyu Tang.

**Writing – review & editing:** Tianjuan Xia, Yi Min.

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
