## [Decision Letter · Decision Letter 0]

13 Aug 2025

Dear Dr. Xia,

Thank you for submitting your manuscript to PLOS ONE. After careful consideration, we feel that it has merit but does not fully meet PLOS ONE’s publication criteria as it currently stands. Therefore, we invite you to submit a revised version of the manuscript that addresses the points raised during the review process.

Please submit your revised manuscript by Sep 27 2025 11:59PM. If you will need more time than this to complete your revisions, please reply to this message or contact the journal office at plosone@plos.org. . . . Please include the following items when submitting your revised manuscript:

We look forward to receiving your revised manuscript.

Kind regards,

Rawaa A. Faris, Ph.D.

Academic Editor

PLOS ONE

https://journals.plos.org/plosone/s/file?id=ba62/PLOSOne_formatting_sample_title_authors_affiliations.pdf....

“This research was financially supported by the Wuhan University Clinical Nursing Special Research and Cultivation Fund Project (1607/600400021)”

“This research was financially supported by the Wuhan University Clinical Nursing Special Research and Cultivation Fund Project (1607/600400021)”

“This research was financially supported by the Wuhan University Clinical Nursing Special Research and Cultivation Fund Project (1607/600400021). The authors thank Liangyan Li for providing professional training and guidance on the operation of the ATP detection equipment and Chao Wang for participating in some data analysis.”

“This research was financially supported by the Wuhan University Clinical Nursing Special Research and Cultivation Fund Project (1607/600400021)”

7. Please remove your figures from within your manuscript file, leaving only the individual TIFF/EPS image files, uploaded separately. These will be automatically included in the reviewers’ PDF.

8. Please review your reference list to ensure that it is complete and correct. If you have cited papers that have been retracted, please include the rationale for doing so in the manuscript text, or remove these references and replace them with relevant current references. Any changes to the reference list should be mentioned in the rebuttal letter that accompanies your revised manuscript. If you need to cite a retracted article, indicate the article’s retracted status in the References list and also include a citation and full reference for the retraction notice

Reviewers' comments:

Reviewer's Responses to Questions

**Comments to the Author**

1. Is the manuscript technically sound, and do the data support the conclusions?

Reviewer #1: Yes

Reviewer #2: Yes

2. Has the statistical analysis been performed appropriately and rigorously?

Reviewer #1: Yes

Reviewer #2: Yes

3. Have the authors made all data underlying the findings in their manuscript fully available?

Reviewer #1: Yes

Reviewer #2: Yes

4. Is the manuscript presented in an intelligible fashion and written in standard English?

Reviewer #1: Yes

Reviewer #2: Yes

Reviewer #1: • Authors should check for writing and typing errors.

• The statements in discussion are acceptable but few paragraphs about the justification of your findings and comparison with other recent relevant studies.

• Only include current up to date references in the reference list and remove outdated ones.

Reviewer #2: • The manuscript within the scope of the journal.

• Both the quality and data presentation of this manuscript are acceptable and of great importance to clinicians and even patients.

• The manuscript expands our knowledge about Effective cleaning of reusable dental instruments

• The title should be revised and reduced its characters.

• The abstract should reflect the content of the article and must be with range of 250-300 words.

• Four to six keywords representing the main content of the article BUT not mentioned in the title.

• More paragraphs should be incorporated to introduction and discussion although the statements in discussion are acceptable but few paragraphs about the justification of your findings and comparison with other recent relevant studies.

• All numerical values and percentage should be omitted from conclusion section.

• The author( s) should pay attention to writing and typing errors throughout the manuscript

• Up to date references should be kept in your reference list and the old should be omitted.

Good Luck

.

Reviewer #1: **Yes:** Afrah AldelaimiAfrah AldelaimiAfrah AldelaimiAfrah Aldelaimi

Reviewer #2: **Yes:** Tahrir N AldelaimiTahrir N AldelaimiTahrir N AldelaimiTahrir N Aldelaimi

While revising your submission, please upload your figure files to the Preflight Analysis and Conversion Engine (PACE) digital diagnostic tool, https://pacev2.apexcovantage.com/. PACE helps ensure that figures meet PLOS requirements. To use PACE, you must first register as a user. Registration is free. Then, login and navigate to the UPLOAD tab, where you will find detailed instructions on how to use the tool. If you encounter any issues or have any questions when using PACE, please email PLOS at . PACE helps ensure that figures meet PLOS requirements. To use PACE, you must first register as a user. Registration is free. Then, login and navigate to the UPLOAD tab, where you will find detailed instructions on how to use the tool. If you encounter any issues or have any questions when using PACE, please email PLOS at . PACE helps ensure that figures meet PLOS requirements. To use PACE, you must first register as a user. Registration is free. Then, login and navigate to the UPLOAD tab, where you will find detailed instructions on how to use the tool. If you encounter any issues or have any questions when using PACE, please email PLOS at . PACE helps ensure that figures meet PLOS requirements. To use PACE, you must first register as a user. Registration is free. Then, login and navigate to the UPLOAD tab, where you will find detailed instructions on how to use the tool. If you encounter any issues or have any questions when using PACE, please email PLOS at figures@plos.org. Please note that Supporting Information files do not need this step.. Please note that Supporting Information files do not need this step.

---

## [Author Response · Author response to Decision Letter 1]

8 Sep 2025

We would like to express our sincere thanks to the reviewers for the constructive and positive comments. The detailed responses to the reviewers' comments are as follows.

According to the requirements of the journal, we have made corresponding checks and made the following revisions.

1. Our manuscript was completed in accordance with the formatting requirements of PLOS ONE's style.

2. The amended Funding Statement was added in cover letter.

3. The Wuhan University Clinical Nursing Special Research and Cultivation Fund Project (1607/600400021) was the only funder. The funder had no role in study design, data collection and analysis, decision to publish, or preparation of the manuscript.

4. All raw data required to replicate the results of our study was uploaded as supporting information in the file named “Data”.

5. I have an ORCID iD (0009-0002-789-9798), and also updated my information.

6. According to the requirements, we have removed the funding information in the Acknowledgments Section.

7. All figures were removed from revised manuscript file, and separately uploaded the individual TIFF/EPS image files.

8. Our reference was reviewed and it is complete and correct. No retracted paper was cited in our paper.

Replies to Reviewer #1

• Authors should check for writing and typing errors.

--According to the comment raised by the reviewer, we had checked the spelling and grammar in writing, and highlight with yellow.

• The statements in discussion are acceptable but few paragraphs about the justification of your findings and comparison with other recent relevant studies.

--We made corresponding modifications to the discussion and references section. We added two new paragraphs form line 276 (Page 10) to line 298 (Page 11). Minor enhancement was made from line 264 – 267 in Page 10. The integration was added from line 323 – 324 in Page 12.

• Only include current up to date references in the reference list and remove outdated ones.

--We deleted some outdated references (number 5, 8,17 and 25 in the original version), and added some current references (number 22 - 29 in revised manuscript and marked with red).

Replies to Reviewer #2

• The title should be revised and reduced its characters.

--We revised and reduced the title characters and marked with red.

• Four to six keywords representing the main content of the article BUT not mentioned in the title.

-- We have revised some keywords and marked with red.

• More paragraphs should be incorporated to introduction and discussion although the statements in discussion are acceptable but few paragraphs about the justification of your findings and comparison with other recent relevant studies.

-- This opinion is similar to the second question raised by Reviewer #1. It has already been addressed previously.

• All numerical values and percentage should be omitted from conclusion section.

--There was no numerical values and percentage in conclusion section.

• The author should pay attention to writing and typing errors throughout the manuscript.

-- we had checked the spelling and grammar in writing, and highlight with yellow.

• Up to date references should be kept in your reference list and the old should be omitted.

--According to the comment raised by the reviewer, we had deleted the outdated references.

---

## [Decision Letter · Decision Letter 1]

8 Oct 2025

Dear Dr. Tianjuan Xia,

Thank you for submitting your manuscript to PLOS ONE. After careful consideration, we feel that it has merit but does not fully meet PLOS ONE’s publication criteria as it currently stands. Therefore, we invite you to submit a revised version of the manuscript that addresses the points raised during the review process.

We look forward to receiving your revised manuscript.

Kind regards,

Rawaa A. Faris, Ph.D.

Academic Editor

PLOS ONE
---

## [Author Response · Author response to Decision Letter 2]

27 Oct 2025

Dear Ph.D. Rawaa A. Faris，

We would like to express our sincere thanks to your opinion about our manuscript. The detailed responses to your comments are as follows.

--- The reviewers did not recommend any papers for citation.

---The references were checked and revised. The revised references were marked with red.

---The reference [5] was updated. The old one “Sterilization of health products in public services (2016)” was replaced by the new one “Australian Dental Association (2024) Guidelines for Infection Control (2024). Fifth Edition”.

--- The old title was wrong in the reference [18]. The corrected title “The Role of ATP Luminometers in Infection Control” was marked with red.

---The time was wrong in the reference [19]. The correct time is 2013.

---No retracted paper was cited in our manuscript.

---The PACE corrected figures were obtained through the Preflight Analysis and Conversion Engine (PACE) digital diagnostic tool, and re-uploaded.

Sincerely

Tianjuan Xia

---

## [Decision Letter · Decision Letter 2]

13 Apr 2026

Impact of Moisturizing Pretreatments on Small Reusable Dental Instruments Cleanliness and Mechanics

PONE-D-25-27990R2

Dear Dr.Tianjuan Xia ,

We’re pleased to inform you that your manuscript has been judged scientifically suitable for publication and will be formally accepted for publication once it meets all outstanding technical requirements.

Kind regards,

Rawaa A. Faris, Ph.D.

Academic Editor

PLOS One

Additional Editor Comments (optional):

Dear Authors,

Following the submission of your revised manuscript entitled:

“Impact of Moisturizing Pretreatments on Small Reusable Dental Instruments Cleanliness and Mechanics”

I am pleased to inform you that, after careful evaluation of the revised version and your detailed responses to the reviewers’ comments, your manuscript has been accepted for publication.

Regards,

Rawaa A. Faris

---

## [Editor Report · Acceptance letter]

PONE-D-25-27990R2

PLOS One

Dear Dr. Xia,

I'm pleased to inform you that your manuscript has been deemed suitable for publication in PLOS One. Congratulations! Your manuscript is now being handed over to our production team.

Kind regards,

on behalf of

Dr. Rawaa A. Faris

Academic Editor

PLOS One